# Theoretical Study of Copper Squarate as a Promising Adsorbent for Small Gases Pollutants

**DOI:** 10.3390/molecules29133140

**Published:** 2024-07-02

**Authors:** Celia Adjal, Nabila Guechtouli, Vicente Timón, Francisco Colmenero, Dalila Hammoutène

**Affiliations:** 1Laboratory of Thermodynamics and Molecular Modeling, Faculty of Chemistry, University of Science and Technology Houari Boumediene (USTHB), BP32, El Alia, Bab Ezzouar, Algiers 16111, Algeria; adjalcelia5@gmail.com (C.A.); n.guechtouli@univ-boumerdes.dz (N.G.); dhammoutene@yahoo.fr (D.H.); 2Instituto de Estructura de la Materia, CSIC, Serrano 121, 28006 Madrid, Spain; 3Faculty of Sciences, University of M’hamed Bougara, (UMBB), Boumerdes 35000, Algeria; 4Centro de Investigaciones Energéticas, Medioambientales y Tecnológicas (CIEMAT), Avda/Complutense, 40, 28040 Madrid, Spain; francolm@ucm.es

**Keywords:** copper squarate, metal–organic framework, molecular adsorption, DFT, adsorption isotherms

## Abstract

Copper squarate is a metal–organic framework with an oxo-carbonic anion organic linker and a doubly charged metal mode. Its structure features large channels that facilitate the adsorption of relatively small molecules. This study focuses on exploring the potential of adsorbing small pollutants, primarily greenhouse gases, with additional investigations conducted on larger pollutants. The objective is to comprehend the efficacy of this new material in single and multiple molecular adsorption processes using theoretical methods based on density functional theory. Furthermore, we find that the molecular adsorption energies range from 3.4 KJ∙mol^−1^ to 63.32 KJ∙mol^−1^ depending on the size and number of adsorbed molecules. An exception is noted with an unfavorable adsorption energy value of 47.94 KJ∙mol^−1^ for 4-nitrophenol. More importantly, we demonstrate that water exerts an inhibitory effect on the adsorption of these pollutants, distinguishing copper squarate as a rare MOF with hydrophilic properties. The Connolly surface was estimated to give a more accurate idea of the volume and surface accessibility of copper squarate. Finally, using Monte Carlo simulations, we present a study of adsorption isotherms for individual molecules and molecules mixed with water. Our results point out that copper squarate is an efficient adsorbent for small molecular pollutants and greenhouse gases.

## 1. Introduction

Nowadays, scientists around the world are actively seeking new porous materials to tackle the pollution crisis caused by urbanization and emissions from motor vehicles and industry [1,2]. Extensive research has been conducted, resulting in the publication of in-depth studies on air emissions and the various types of pollutants present in our atmosphere, such as carbon dioxide (CO_2_), the main contributor to the greenhouse effect, which is largely responsible for global warming. Additionally, methane (CH_4_), tropospheric ozone (O_3_), dinitrogen monoxide (N_2_O), nitrogen trifluoride (NF_3_), and sulfur hexafluoride (SF_6_) are critical trace gases that contribute to climate change. These six compounds are among the main greenhouse gases identified in the Kyoto Protocol [3,4,5,6].

In addition, the combustion of coal, fossil fuels, or biomass also releases air pollutants, such as sulfur dioxide (SO_2_) and volatile organic compounds (VOCs), which contribute significantly to industrial and vehicle exhaust pollution. At the same time, VOCs produce a variety of pollutants, including polycyclic aromatic hydrocarbons (PAHs) derivatives such as NPAH nitrates. This category includes, for example, 4-nitrophenol (4NP), which is considered one of the most hazardous pollutants in our daily lives [7,8,9,10,11,12,13,14,15]. Nitrophenols are classified as priority pollutants by the U.S. Environmental Protection Agency [16]. Due to their high toxicity, atmospheric pollutants have considerable negative effects on human health, wildlife, and global warming. For this reason, emphasis has been placed on experimental and theoretical studies concerning the application of separation, adsorption, gas capture and storage, and selective catalysis as potential solutions [17,18,19,20,21,22].

Recently, metal–organic frameworks (MOFs) have been in great demand for the capture and storage of gaseous pollutants due to their extraordinary porosity, surface area, crystallinity, controllable structure, and chemical separation capabilities [23,24,25]. Consequently, a plethora of work has been carried out on various materials, such as zeolitic imidazolate frameworks (ZIFs), clay minerals, lignite, anthracites, and more, to adsorb pollutants [26,27,28,29,30]. The presence of water has also been considered, providing valuable information on its impact on the adsorption process [31,32,33].

In this paper, we will focus on a type of MOF [34] that has not yet been explored for adsorption or capture processes: the copper squarate (C-S) metal–organic framework. The crystal structure of this material was first analyzed and described in detail by Dinnebier et al. [35,36]. C-S was subsequently investigated by Colmenero et al. [37] using theoretical solid-state methods based on density functional theory (DFT). This study firmly established the crystal structure of copper squarate, previously determined using a restricted rigid-body Rietveld refinement, which discovered that it exhibits the phenomenon of negative linear compressibility due to the presence of the empty channel structural motif. Colmenero et al. demonstrated that other squarate MOFs, such as zinc [36] and uranyl squarates [38], also exhibit negative linear compressibility. In fact, Colmenero [39] has shown that oxo-carbonic acids in the solid state exhibit extremely anomalous mechanical behavior, and recently, Qiu et al. [40] have revealed that lead and barium squarates exhibit zero linear compressibility and thermal expansion effects over a wide range of pressures. The presence of empty spaces inside these structures and their large pore size have led us to believe in the viability of these materials for the capture and storage of hazardous gases. As far as we know, squarate MOFs have only been recently studied for molecular adsorption applications. Namely, the use of cobalt squarate MOF has recently been investigated for CO_2_/N_2_ separation by Zhang et al. [41].

The aim of our work is to characterize the behavior of squarate materials in molecular adsorption applications, which have not yet been explored, using solid simulation techniques. Specifically, we will investigate the ability of copper squarate to absorb a series of different classes of gaseous pollutants in the presence and absence of water. We will closely examine its response to these pollutants, determining its preferred adsorption sites, energies, and mechanisms, as well as its adsorption isotherms. Unfortunately, to our knowledge, no experimental or theoretical studies have been published on adsorption in copper squarate. The present work is therefore the first to provide theoretical results that will serve as a starting point for further experiments. The exploration of new materials for the adsorption of contaminants is crucial to counter the wave of pollution caused by urbanization that we are currently experiencing.

## 2. Results and Discussion

The copper squarate crystal structure was re-optimized, and the results obtained are fairly consistent with those obtained by Colmenero et al. [36]. The powder X-ray diffraction (PXRD) patterns of copper squarate were derived from the computed and experimental structures using the program REFLEX included in the Materials Studio program suite [42]. The results are displayed in Appendix A and, as can be seen, they are consistent. Because of the small size of the unit cell, it was doubled along the axis to facilitate the adsorption of the pollutant, as was conducted in previous works studying the adsorption of similar molecules in materials like zeolite [43]. The calculations were then continued with a series of optimizations of the double unit cell containing the pollutant gases. These gases were introduced inside using the Adsorption Locator tool of Materials Studio [42] in order to conduct the adsorption processes inside this material. The results are presented in Table 1, where the lattice parameters of the crystal unit and the double cell of copper squarate are provided, as well as the parameters of the cells distorted due to the capture of one or more pollutants and water molecules. The optimized geometries of different species are compared with the corresponding experimental data in Appendix A. The optimized structure of copper squarate is shown in Figure 1. The optimized copper squarate unit and double cell can be downloaded from the Appendix A).

### 2.1. Crystal Bulk Structure and Analysis of Connolly Free Volume and Surface Area Properties

In this section, we employ another feature of Materials Studio 22.0 [42], the “Atoms Volume and Surfaces” to approximate the Connolly surface of copper squarate and the pollutants separately using the probe radius. This allows us to thoroughly explore the cavity and surface features of the internal complex structure after the adsorption process [44] and to obtain the occupied and free volumes, as well as surface area (Table 2). These parameters are then used to calculate the pore volume using the expression below [45]:(1)$$Pore\;volume\;{(cm}^{3}\cdot g^{-1})=\frac{Cell\;Free\;Volume\;\left({Å}^{3}\right)}{Density\;{(g\cdot cm}^{-3})\times Cell\;Volume\;({Å}^{3})\;}$$

From Table 2, we notice a good agreement between the experimental density (2.427 $g\cdot cm^{-3}$) and the theoretical value (2.263 $g\cdot cm^{-3}$), showing the accuracy of the DFT-D3 method in describing unit cell structures in crystalline materials. The total pore volume of copper squarate (1 × 1 × 2 cell) is expected to be 0.0958 $g\cdot cm^{-3}$, i.e., 23% of the total cell volume reported in Table 1. This value is close to the experimental one (24.6%) approximated from available data (density and cell volume) in the work of Colmenero et al. [36] considering the (1 × 1 × 2) cell, as well as using the theoretical cell free volume. All these calculations use a probe radius of 1.0 Å and a grid interval of 0.6 Å [45].

Moreover, we employed the same grid interval of 0.6 Å for the Connolly surface parameters to estimate the volume and surface area of other adsorbed species. Meanwhile, we varied the probe radius from one molecule to another due to the electronic density of those molecules [32,45]. For H_2_O, SO_2_, N_2_O, and CH_4_, we use a probe radius of 1.37 Å due to their similar electronic cloud characteristic except for CH_4_. The corresponding results were (18.99 Å^3^/34.79 Å^2^), (40.48 Å^3^/58.08 Å^2^), (28.87 Å^3^/50.42 Å^2^), and (27.47 Å^3^/46.18 Å^2^), respectively. In the case of 4NP, a slightly larger probe radius of 1.4 Å was employed to achieve a result of (115 Å^3^/138.02 Å^2^). Finally, for the more electronegative molecules CO_2_, O_3_, NF_3_, SF_6_, CF_4_, and 2NP, a probe radius of 1.7 Å was chosen, considering their higher electron density due to hindrance, resulting in (32.28 Å^3^/51.37 Å^2^), (31.25 Å^3^/50.37 Å^2^), (40.71 Å^3^/60.59 Å^2^), (74.81 Å^3^/89.45 Å^2^), (50.04 Å^3^/69.05 Å^2^), and (110.49 Å^3^/134.58 Å^2^), respectively. These results indicate the ability of C-S to adsorb the pollutants listed, except 2NP and 4NP, which have almost the same Connolly surface and volume as the filter, potentially causing repulsive disturbances.

To delve deeper into the understanding of C-S capacity as a filter and elucidate the outcomes of the Connolly analysis, we employed the DFT-D3 veracity method. This method allows us to ascertain the maximum quantity of adsorbed pollutants within the bulk. Table 3 shows the structural parameters of the co-adsorption process.

### 2.2. Adsorption Analysis

Achieving optimal DFT performance in this study involves a hierarchical approach. First, we locate the most stable configuration in conformational space, as was previously described using the Adsorption Locator module with COMPASS III force field [46], by dropping the molecule at various pressures employing the Metropolis algorithm. The DFT geometry is then optimized for the most favorable configuration. (Figure 2) shows the single final adsorption structure of pollutants and water in C-S and (Table 4) gives the energy data for the adsorption process. Negative adsorption energies indicate favorable capture of gases [31].

#### 2.2.1. Pollutants and Water Adsorption

The results indicate favorable adsorption of CO_2_, N_2_O, SO_2_, and H_2_O molecules into C-S using force field loading, with up to five, one, one, and eighteen species per double unit cell, respectively. The accuracy of these findings was validated by CASTEP calculations, revealing adsorption energies of −18.93, −63.32, −38.04, and −51.16 $KJ\cdot {mol}^{-1}$, respectively, for a single captured molecule. This suggests the existence of electrostatic interactions, as illustrated in the zoomed images in Figure 2. For all four species, we observe a short consistent interaction involving C-S oxygen with C (CO_2_), N (N_2_O), S (SO_2_), and strong hydrogen [47] bound with H (H_2_O), comprising a bound distance of 2.811, 2.954, 2.775, and 1.966 Å, respectively. In addition, a weaker interaction is evident with C (C-S), interacting with O (CO_2_), N (N_2_O), O (SO_2_), and O (H_2_O) at distances ranging from 2.847–3.080 Å.

In Table 4, the calculated energies are CH_4_ (−219.678 eV), CO_2_ (−1016.651 eV), O_3_ (−1285.930 eV), SF_6_ (−4198.029 eV), NF_3_ (−2218.200 eV), CF_4_ (−2751.691 eV), N_2_O (−979.812 eV), SO_2_ (−1165.160 eV), H_2_O (−463.290 eV), 4NP (−2574.182 eV), and C-S (−28,157.254 eV), referring to the number of adsorbed pollutant and water molecules.

Generally, metal–organic frameworks (MOFs) are known to be unstable in humid environments. Zeolite has been highlighted in the literature for its hydrophobic nature [48,49,50]. However, recent experimental studies have identified exceptions, such as ZIF-8 and ZIF-90, which show stability in the presence of water [51,52]. In our study, we theoretically demonstrate that C-S exhibits stability in the presence of water, which is consistent with the hydrophilic behavior observed in some MOFs. In particular, we show that O-H bond lengths in water approach 0.966 Å, which closely matches the experimental value of 0.98 Å [52].

Methane is a non-polar molecule and has been the subject of previous adsorption studies, particularly on zeolites [45]. In our study, different quantities of CH_4_ ranging from one to four were absorbed and optimized to identify the most favorable insertion sites. Our results are consistent with those of a previous study in which four methane molecules were observed inside the ZIF-4 cell [32]. By analyzing the modeled structures, we observed that all CH_4_ molecules are in close proximity to the C-S, involving a hydrogen interaction with the oxygen atom of C-S, with a bonding distance of 2.737 Å, as illustrated in (Figure 2). This interaction represents a limiting case of hydrogen bonding, characterized by a weak donor to a strong acceptor due to their substantial contribution to dispersion [47]. This contributes to system stabilization, resulting in negative adsorption values ranging from −19.42 $KJ\cdot {mol}^{-1}$ to −23.02 $KJ\cdot {mol}^{-1}$ for one and four CH_4_ uptake molecules, respectively.

From Table 4, we can observe high adsorption energies of −30.51, −50.05, and −26.94 $KJ\cdot {mol}^{-1}$ for the adsorption of one, three, and five O_3_ molecules, respectively, including a strong uptake within the bulk. This is attributed to the reactive nature and high polarity of ozone due to the presence of oxygen atoms. Capture involves intermolecular Van der Waals forces, mainly O_O3_-O_C-S_ with an average bond length of 2.8 Å for all the optimized structures. In this case, the charged fluctuation of the oxygen and the electronic density leads to a sensitive instantaneous dipole moment, creating a type of London dispersion.

The vacuum inside the cavity allows the capture of one molecule with fluorine atoms due to the steric effect of the electronic density of the halogen, causing a repulsion with the oxo-carbonic anion of the copper squarate. We observed that the adsorption energy is −14.12 $KJ\cdot {mol}^{-1}$ for NF_3_, while it decreases consistently for CF_4_ and SF_6_, with values of −3.40 and −3.78 $KJ\cdot {mol}^{-1}$, respectively. We deduce that as the number of fluorine atoms increases, the adsorption energy decreases. Additionally, we notice that when the central atom is nitrogen, the adsorption energy is higher, as seen with N_2_O. The optimized lengths are shown in Figure 2, where it can be seen that the carbon of C-S interacts with the fluorine of CF_4_ and NF_3_ at 2.7 and 2.8 Å, while a shorter value of 1.98 Å is noted with SF_6_ because its octahedral form minimizes the distance between the pollutant and the bulk.

A positive adsorption energy of +47.94 $KJ\cdot {mol}^{-1}$ was noted for 4NP (Table 4), indicating unfavorable capture of the aromatic pollutant by the C-S. This is in contradiction with the adsorption locator calculation, which allows the uptake of one molecule. This implies the lack of accuracy of the force field and the foremost importance of using the DFT-D3 method to complement and achieve a reliable calculation.

#### 2.2.2. Co-Adsorption of Pollutants and Water

In this section, we study the effect of a humid environment on the adsorption inside copper squarate bulk. The adsorption energy and interaction energy are calculated using Equations (2) and (3), respectively, and the results are presented in Table 4.

According to Table 4, the co-adsorption of the pollutant with water gives negative adsorption energy, revealing favorable adsorption energy, except for 4NP. The latter has a positive adsorption value in both dry and humid environments, which can be caused by the unavailability of sufficient space on the pores due to its large size and radius.

For CH_4_, SO_2_, and CO_2_, the adsorption energies increased in the presence of water to reach −33.11, −43.48, and −40.39 $KJ\cdot {mol}^{-1}$, respectively. This implies the same interaction as above between the pollutant and the bulk. In addition, weak hydrogen bonds are formed between the water and the pollutant, which further increases the adsorption energy. The same trend is noted for molecules with fluorine atoms, where the values of the mixed adsorption energy increased to reach −26.83, −41.31, and −6.186 $KJ\cdot {mol}^{-1}$ for NF_3_, CF_4_, and SF_6_, respectively. The important impact on the adsorption energetic value is due to the multiple interactions caused by the reactive sites of these types of pollutants. In contrast, for ozone and dinitrogen monoxide, the adsorption energy decreases slightly but remains favorable to the adsorption process. Most of the interaction energies are positive, indicating repulsion between the species and far adsorption between the small pollutants and the water inside the pore. This is due to the electron density, as depicted in Figure 3.

Finally, the adsorption of 4NP in the presence of water does not counterbalance capture towards a favorable process. In previous work on nitrophenol on activated carbon bulk under humid conditions, the authors observed a possibility of adsorption by water as it moved to place itself between 4NP and the surface bulk, creating hydrogen bonds and electrostatic interactions. This suggests a lack of space inside the C-S pore, which can create repulsion due to the high charge density of water, 4NP, and C-S cell, as can be seen in Figure 3 where water is placed in the cell extremity, not inside. We also noticed this in the previous section on the Connolly free volume, where we noted that 4NP and 2NP have almost the same volume. The interaction energy here is negative due to the hydrogen bonding of the aromatic compound to the copper squarate.

These fluctuations in no way interfere with copper squarate’s favorable ability to capture these small pollutants. On the contrary, they positively highlight the fact that the adsorbent can capture them in both dry and humid states, without the presence of water playing a counterbalancing role. It is important to remember that the small pollutants, as well as water, are energetically stable separately in the C-S.

The main geometric parameters of the co-adsorption (water/pollutant) were analyzed and are reported in Table 5 above. The results indicate that there is no significant change in the bond distance and valence angle between them compared with the isolated adsorbed form. This is due to the presence of more free space, allowing more water molecules to be captured, confirming the observation that the interaction energy is positive. The presence of additional adsorbed molecules can lead to an increase in distortion due to space limitations. This phenomenon has already been observed in solid simulations, notably in studies of methane, where the angle of increased from 109° to 114° [45].

### 2.3. Sorption Isotherms

The isothermal sorption of pollutants and water in singular and mixed forms was studied at 298 K using the “sorption isotherm” module implemented in Materials Studio 22.0 [42]. This study is based on random moves to perform energy changes for the new conformation. This is by measuring the chemical potential, expressed as the gas fugacity with pressure inside the cell bulk. The results are shown in Figure 4, indicating the average adsorption capacity of gas molecules by the copper squarate.

By analyzing the adsorbed singular graphs in Figure 4, we observe that most of the pollutants present a consistent loading profile with molecules inserted around zero or zero per the considered unit cell in low- or high-pressure variations. For example, N_2_O shows zero loaded molecules, while the adsorption study shows high negative adsorption energy of −63.32 kJ/mol, indicating a favorable capture. This aligns also with the Connolly studies, which show available space inside copper squarate. These particular results contradict the precise results obtained by the DFT-D3 method. These strange results can be attributed to the limitations of the force field used in this C-S model, which includes pressure and is unable to accurately describe the unusual properties of the material, in particular its negative linear compressibility [36], highlighting the necessity of complementing it with ab initio calculations. In contrast, water is well described, showing an uptake loading of 15 molecules per cell, which is in good agreement with CASTEP calculations. This agreement may indicate the established hydrophilic nature of the solid. Similarly, ozone molecules show comparable behavior due to their common nature with water.

In the presence of water, the line graph representing the pollutant does not change. There are two reasons for this. The first reason is the same as when they are loaded individually. The second reason may be due to the hydrophilicity of water, which dominates the pollutants and competes with them, reducing their loading capacity in the cell. This phenomenon is evident with ozone, which loads around three molecules instead of six in a dry environment. This is due to the stronger hydrogen bonds between the water and the copper squarate to the intermolecular interactions between the pollutants and the bulk.

## 3. Quantum Computational Methods

All calculations for the solid-state compounds considered in this work were performed with CASTEP code [53] using theoretical methods based on DFT using norm-conserving pseudopotentials and plane wave basis sets [54]. Due to the large pore size of MOFs, which contain hundreds of atoms or more, quantum mechanical methods such as post-HF are impractical for such large systems [55]. The geometry optimizations were performed employing the Broyden–Fletcher–Goldfarb–Shanno scheme (BFGS) [55,56]. We treated the exchange-correlation energy term using the generalized gradient approximation (GGA) with Perdew–Burke–Ernzerhof (PBE) density functionals [57]. Since the pollutant molecules are packed together inside the unit cell of copper squarate, their description requires the inclusion of Van der Waals forces. For this purpose, Grimme’s empirical dispersion correction method (DFT-D3) for dispersive vdW interactions was used in calculations [58,59]. Customized atom pseudopotentials [60] generated with the CASTEP code and a plane wave kinetic energy cut-off of 1000 eV were utilized. Convergence criteria were set at 2 × 10^−5^ eV/atom for energy, 0.05 eV/Å for interatomic forces, 0.1 GPa for maximum stress, and 0.002 Å for displacements. Atomic forces and charges were evaluated at the minimum in potential energy surface to predict the harmonic vibrational infrared spectrum using density functional perturbation theory [61,62].

After a full geometry optimization of all CH_4_, CO_2_, O_3_, SF_6_, NF_3_, CF_4_, N_2_O, SO_2_, 4NP, water, and copper squarate, the most stable adsorption sites of the pollutants inside the solid material were determined in order to minimize the first adsorption energy. The adsorption sites and mechanisms within the copper squarate were determined by Monte Carlo simulations [63], implemented in the Adsorption Locator module using COMPASS III [46] force field in Materials Studio 22.0 [42] code. Simulations involve a fraction of 10,000 steps following a number of cycles of 3 with 15,000 steps per cycle. Subsequent to the initial results, Density Functional Theory (DFT) calculations were conducted to optimize performance by adjusting the generic force fields within the Adsorption Locator module. More specifically, it was already proved that force fields such as COMPASS III [46] and Dreiding [64] tend to overestimate the number of molecules adsorbed in simulations [65], as was demonstrated in previous studies involving ZIF-4 or ZIF-6 [32,45].

The adsorption process occurs through chemical or physical binding, facilitated by the attractive forces between the solid cavity of the copper squarate and the pollutants, in both dry and humid conditions. The adsorption energies were calculated according to equations [66]:(2)$${∆E}_{ads}(X)=\left[E_{T}-\left(E_{C-S}+nE_{X}\right)\right]/n\left(X=Polluants\;or\;H_{2}O\right)$$
(3)$${∆E}_{ads}^{mult}\left(X,Y,\cdots \right)=\frac{\left[E_{T}-\left(E_{C-S}+nE_{X}+mE_{Y}+\cdots +tE_{Z}\right)\right]}{\left(n+m+\cdots +t\right)}/\left(X,Y,Z\cdots =Polluants\;or\;H_{2}O\right)$$

In these equations, ${∆E}_{ads}$ refers to the isolated adsorption energy for each individual species (X) and ${\;{∆E}_{ads}^{mult}}_{\;}$ denotes the mixed adsorption energy for species (X,Y, ⋯…). $E_{T}$ stands for the total energy of copper squarate (C-S) with gas molecules included, $E_{C-S}$ corresponds to the energies of pure C-S, and $E_{X}$ is the energy of the respective pollutants. In addition, n, m, t represent the quantities of loaded molecules. The assessment of pollutants and water is conducted using a 10 Å supercell. A negative adsorption energy implies favorable adsorption.

The interaction energies between the co-adsorbed pollutants on the C-S material were calculated with the help of the energetic database obtained from DFT calculations using equation [45]:(4)$$E_{int}=E_{\left(n\right)-pollutants+Bulk}+E_{Bulk}-\sum E_{n-pollutants+Bulk}$$
where $E_{\left(n\right)-pollutants+Bulk}$ represents the total energy of the solid material after adsorption of the n pollutants by copper squarate bulk, and $\sum E_{n-pollutants-Bulk}$ is the sum of the total energy of the previously adsorbed species calculated separately.

The Grand Canonical Monte Carlo (GCMC) method [67] implemented in the Sorption module of Materials Studio 22.0 [42] was used to determine adsorption isotherms with the Metropolis algorithm and COMPASS III. The gas molecules were randomly loaded at a fixed temperature of 298 K into the fixed-volume copper squarate structure based on the gas fugacity. The simulations involve 100,000 equilibration steps followed by 1,000,000 Monte Carlo steps, assuming periodic boundary conditions. Long-range interactions were treated using the Ewald and group summation method with an Ewald accuracy of 0.0001 kcal/mol and a cut-off distance of 15.5 Å. The results are presented as a line graph (N = f (µ)), where f is the density of the number of loaded molecules N, expressed as a function of chemical potential.

## 4. Conclusions

In this work, we focus our interest on a new kind of MOF, namely copper squarate, whose adsorption behavior has not been explored in previous studies. For this reason, the adsorption of a series of small pollutants and some aromatic compounds was investigated using DFT calculations in a dry and humid environment. As a first observation, we note that small pollutants such as CH_4_, CO_2_, O_3_, SF_6_, NF_3_, CF_4_, N_2_O, and SO_2_ favor the adsorption by different amounts depending on their chemical composition.

Squarate demonstrates excellent stability in the presence of the considered pollutants, exhibiting a favorable capture of the latter that results in a negative adsorption energy value. This phenomenon can be attributed to the existence of electrostatic interaction forces between pollutants and squarate. These interactions are particularly pronounced at short distances, occurring between the oxygen of squarate and the carbon atoms, nitrogen, and sulfur in the molecules CO_2_, N_2_O, SO_2_, and H_2_O, respectively. The presence of other weak parallel interactions further reinforces the binding of these molecules of the C-S. Additionally, the non-polarity of CH_4_ gas makes it a favorable insert within squarate, accentuated by a hydrogen bond. Finally, the dipole nature of O_3_ imparts characteristic properties, enabling it to react very selectively as an electrophile. These characteristic properties enable them to react very selectively as an electrophile. This characteristic facilitates its strong adsorption with squarate, driven by the intermolecular forces of Van der Waals and London dispersion.

Pollutants with higher volume and surface density are adsorbed in smaller quantities; for example, molecules with fluorine o nitrogen atoms. Indeed, adsorption in NF_3_, CF_4_, and SF_6_ molecules is influenced by the number of fluorine atoms used and the adapted geometry. The smaller this number, the more the adsorption process is promoted. Furthermore, the more it adopts a complex structure, the more the interactions between pollutants and squarate are minimized. In contrast to all previous gases, nitrophenols are not able to be adsorbed in this bulk because of the small pore size.

The adsorption energy of the molecule containing nitrogen as N_2_O in a dry environment is revelaed to be the highest one (−63.02), even if the maximal insertion is one molecule per cell compared to four insert molecules of CH_4_ with a maximum value of −23.02. This indicates the strong interaction involved in the first adsorption. Globally, the addition of water does not interfere with the adsorption, keeping the adsorption energy negative.

The co-adoption of CH_4_, SO_2_, and CO_2_ with H_2_O implies an increase in adsorption energy. The same trend is observed for molecules containing fluorine atoms, such as NF_3_, CF_4_, and SF_6_. This impact on adsorption energy values is attributed to multiple interactions caused by the reactive sites of these pollutants, as well as the formation of weak bonds between water molecules.

In a humid environment, the same order is found for O_3_ and N_2_O molecules as in the dry state, with a very slight decrease in adsorption energies. Meanwhile, 4NP is an exception to the other pollutants studied. On the other hand, it is characterized by an unfavorable capture on C-S in both humid and dry environments, resulting in positive adsorption energy. This is attributed to its large size and the unavailability of sufficient space in the pore. However, C-S exhibits a high capacity to insert water inside its cavity, revealing its hydrophilic nature. This behavior stands in contrast to enormous MOFs well-documented in the literature for their highly hydrophobic characteristics. Such hydrophilicity is particularly advantageous, especially considering that gas molecules are typically present in humid conditions, allowing us to regard C-S as a revolutionary MOF for adsorption, making it a novel target in this field.

The number of molecules fixed in the C-S decreases from the dry to the wet state, indicating that water molecules compete with pollutants and form strong hydrogen bonds with the squarate, compared to the interactions between pollutants and the bulk.

In future work, it is recommended to simulate a molecular work with a reduced part of copper squarate to understand deeper what is happening inside the cell cavity and comprehend with better precision to adsorption nature and the type of electrostatic interaction, including Van der Waals or hydrogen bonds in both dry and humid environments, as well as to determine the factors that govern this adsorption phenomenon. In addition, to complete this, a molecular dynamics (MD) trajectory will be helpful to understand dynamic adsorption.

## Figures and Tables

**Figure 1 molecules-29-03140-f001:**
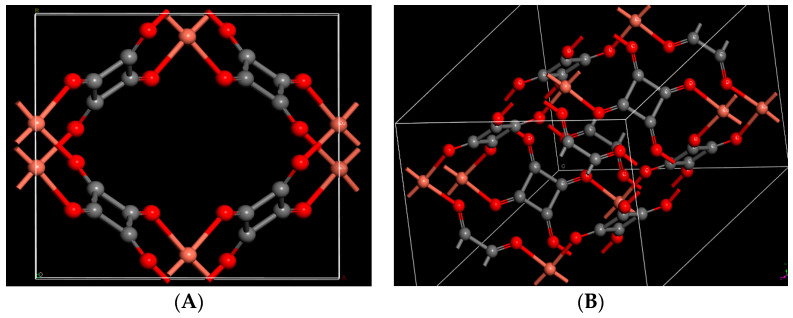
Optimized crystal structure of copper squarate: (**A**) single unit cell; (**B**) double unit cell along edge. (Color code: Red: Oxygen/Orange: Copper/Gray: Carbon).

**Figure 2 molecules-29-03140-f002:**
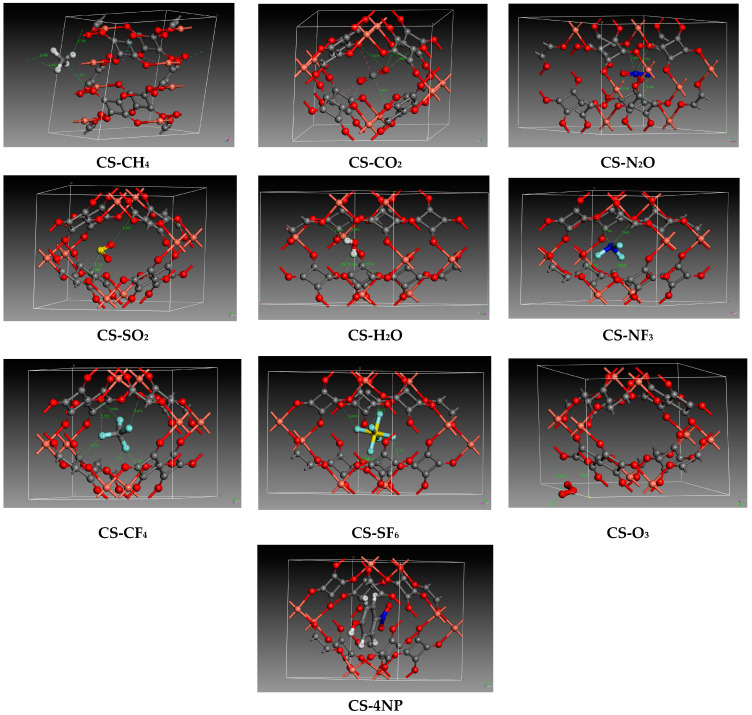
Zoomed interactions of optimized copper squarate containing single pollutants or water. (Color code: Red: Oxygen/Orange: Copper/Gray: Carbon/White: Hydrogen/Royal blue: nitrogen/Yellow: Sulfur/Light blue: Fluorine).

**Figure 3 molecules-29-03140-f003:**
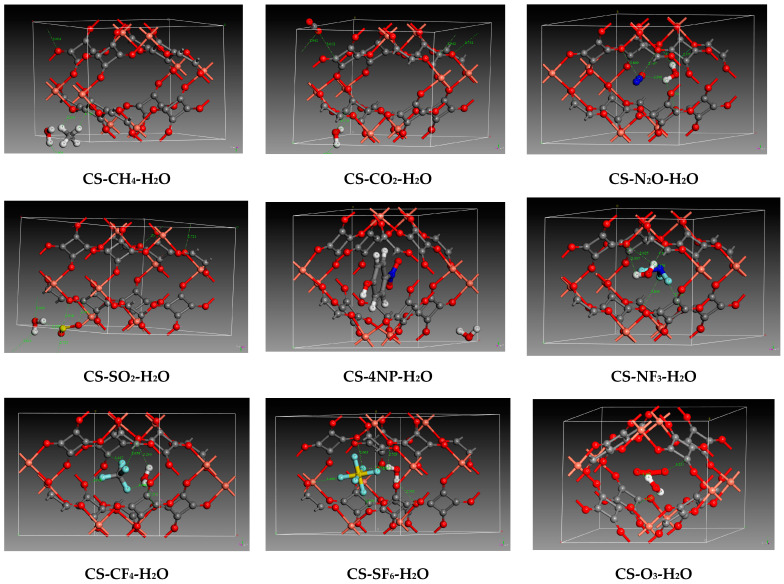
Zoomed interactions of optimized copper squarate containing co-adsorbed pollutants with water. (Color code: Red: Oxygen/Orange: Copper/Gray: Carbon/White: Hydrogen/Royal blue: nitrogen/Yellow: Sulfur/Light blue: Fluorine).

**Figure 4 molecules-29-03140-f004:**
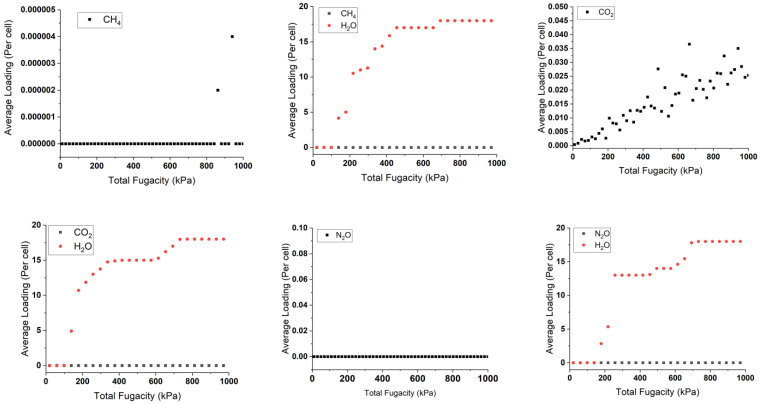
Sorption isotherms of single and mixture adsorbed pollutants with and without water at 298 K.

**Table 1 molecules-29-03140-t001:** Structural parameters of copper squarate (C-S) are in the pure form and contain adsorbed pollutant molecules and water separately. The experimental unit cell parameters are from Ref. [36].

Species	a(Å)	b(Å)	c(Å)	α(°)	β(°)	γ(°)	$\mathrm{Vol}.\;({Å}^{3}$)
C-S (1 × 1 × 2)	11.017	9.352	11.288	89.918	117.636	90.082	1030.39
Exp. (1 × 1 × 1)	10.686	8.939	5.674	90.0	117.600	90.0	480.455
1 CH_4_	11.159	9.145	11.224	89.570	116.579	90.186	1024.40
4 CH_4_	10.707	9.812	11.394	90.089	117.974	89.977	1057.25
1 CO_2_	11.212	9.071	11.208	89.278	116.246	90.607	1022.29
2 CO_2_	11.096	9.264	11.276	89.356	116.571	90.410	1036.68
5 CO_2_	10.926	9.579	11.405	89.330	115.918	89.907	1073.61
1 O_3_	11.062	9.275	11.253	90.081	117.058	89.949	1027.85
3 O_3_	10.972	9.654	11.206	89.975	119.881	90.149	1029.24
5 O_3_	10.937	9.563	11.409	90.103	116.515	90.254	1067.79
1 SF_6_	10.904	9.754	11.226	89.641	118.322	89.846	1051.02
1 NF_3_	11.106	9.225	11.261	90.333	116.512	89.765	1023.45
1 CF_4_	11.015	9.610	11.192	89.801	119.085	90.008	1035.35
1 N_2_O	10.925	9.683	11.217	90.014	120.345	90.007	1024.08
1 S_2_O	11.139	9.167	11.215	90.160	116.500	90.140	1024.94
1 H_2_O	11.214	9.015	11.187	90.279	116.117	89.714	1015.56
5 H_2_O	11.070	9.431	11.155	90.185	117.025	89.544	1037.45
18 H_2_O	10.907	9.524	12.334	93.334	106.045	89.533	1229.24
14NP	10.027	10.51	11.722	88.364	114.868	90.152	1120.26

**Table 2 molecules-29-03140-t002:** Connolly free volume, surface area of supercell sizes, and density of copper squarate.

Sample	$\mathrm{Density}\;(g\cdot cm^{-3})$	$\mathrm{Accessible}\;\mathrm{Surface}\;\mathrm{Area}\;({Å}^{2})$	$\mathrm{Free}\;\mathrm{Volume}\;({Å}^{3})$	$\mathrm{Pore}\;\mathrm{Volume}\;(g\cdot cm^{-3})$
C-S Exp. [36]	2.263 2.427	283.7	223.5	0.0958

**Table 3 molecules-29-03140-t003:** The structural parameters of copper squarate containing a co-adsorbed mixture of a pollutant with water.

Mixture	a(Å)	b(Å)	c(Å)	α(°)	β(°)	γ(°)	$\mathrm{Vol}.\;({Å}^{3})$
1 CH_4_ + 1 H_2_O	11.088	9.246	11.231	89.983	116.856	89.765	1027.29
1 CO_2_ + 1 H_2_O	11.082	9.249	11.216	90.009	116.896	89.695	1025.20
1 O_3_ +1 H_2_O	11.134	9.192	11.229	89.511	116.154	90.203	1030.70
1 SF_6_ +1 H_2_O	10.902	9.796	11.247	90.335	118.839	90.090	1051.76
1 NF_3_ +1 H_2_O	11.021	9.376	11.298	89.924	116.777	89.956	1024.30
1 CF_4_ +1 H_2_O	11.014	9.573	11.187	89.728	118.839	90.084	1033.19
1 N_2_O + 1 H_2_O	11.053	9.533	11.185	90.004	119.398	90.076	1026.75
1 SO_2_ +1 H_2_O	11.095	9.232	11.221	90.499	116.896	89.693	1025.20
1 4NP + 1 H_2_O	9.9750	10.55	11.774	88.224	116.634	90.125	1107.77

**Table 4 molecules-29-03140-t004:** Adsorption and co-adsorption energies in copper squarate calculated with CASTEP.

X	Pollutant	H_2_O	Energy (eV)	${∆E}_{ads}$ $\left(KJ\cdot {mol}^{-1}\right)$	${∆E}_{ads}^{mult}$ $\left(KJ\cdot {mol}^{-1}\right)$	${∆E}_{int}(KJ\cdot {mol}^{-1})$
1	CH_4_	0	−28,377.13	−19.42		
4	CH_4_	0	−29,036.92	−23.02		
1	CO_2_	0	−29,174.10	−18.93		
2	CO_2_	0	−30,190.92	−17.72		
5	CO_2_	0	−33,241.66	−22.22		
1	O_3_	0	−29,443.50	−30.51		
3	O_3_	0	−32,016.10	−50.05		
5	O_3_	0	−34,588.30	−26.94		
1	SF_6_	0	−32,355.32	−3.785		
1	NF_3_	0	−33,375.60	−14.12		
1	CF_4_	0	−30,908.98	−3.400		
1	N_2_O	0	−29,137.72	−63.32		
1	SO_2_	0	−29,322.81	−38.04		
1	H_2_O	0	−28,621.07	−51.16		
5	H_2_O	0	−30,476.14	−47.01		
18	H_2_O	0	−36,505.20	−46.78		
1	4NP	0	−30,730.94	+47.94		
1	CH_4_	1	−28,157.25		−33.11	+4.364
1	CO_2_	1	−29,638.03		−40.39	−10.68
1	O_3_	1	−29,907.21		−35.52	+10.63
1	SF_6_	1	−32,818.70		−6.186	+42.67
1	NF_3_	1	−30,839.30		−26.83	+11.01
1	CF_4_	1	−31,373.09		−41.31	−27.96
1	N_2_O	1	−29,601.17		−39.10	+36.30
1	SO_2_	1	−29,786.61		−43.48	+1.952
1	4NP	1	−30,730.93		+47.93	−10.34

**Table 5 molecules-29-03140-t005:** Main geometrical parameters for the optimized co-adsorption (1 X −1 H_2_O) structures (distances in Å and angles in degrees) on copper squarate X = (CH_4_, CO_2_, O_3_, SF_6_, NF_3_, CF_4_, N_2_O, and SO_2_).

X	H_2_O	d_H-O_		One Distance of (X)	Angles (X)
CH_4_	1	0.961	105.9	C-H = 1.094	109.255
CO_2_	1	0.961	105.7	C-O = 1.165	179.344
O_3_	1	0.961	103.4	O-O = 1.267	117.303
SF_6_	1	0.960	104.8	S-F = 1.584	89.787
NF_3_	1	0.969	104.0	N-F = 1.410	102.390
CF_4_	1	0.961	106.6	C-F = 1.345	109.673
N_2_O	1	0.961	105.9	N-N = 1.140 N-O = 1.149	179.782
SO_2_	1	0.962	103.8	S-O = 1.440	118.205

## Data Availability

The original contributions presented in the study are included in the article and Appendix A; further inquiries can be directed to the corresponding authors.

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
