# Peer review of "Theoretical Study of Copper Squarate as a Promising Adsorbent for Small Gases Pollutants"

_molecules, 2024, doi:10.3390/molecules29133140_

Round 1
Reviewer 1 Report
Comments and Suggestions for Authors
In this manuscript, Vicente and coworkers investigate the copper squarate to absorb a series of small pollutants and some aromatic compounds using DFT calculations in a dry and humid environment. This work demonstrate that water exerts an inhibitory effect on the adsorption of these pollutants, distinguishing copper squarate as a rare MOFs with hydrophilic properties. I suggest this manuscript to be considered as an article in Molecules after taking the following into account:
1. On page 3, lines 105-107, the authors mention that a complete geometry optimization was performed for gas molecules, water, and copper squarate. It is recommended that the charge profiles of the optimized gas model and water molecule model, as well as data bond lengths and angles, compared to the reported experimental values. This comparison should be included in the supporting information.
2. On page 3, lines 109-113, the authors indicate that Monte Carlo simulations were employed to identify the gas adsorption sites and mechanisms within the structure, utilizing the adsorption locator module in Materials Studio. However, they do not provide details regarding the specific settings utilized, including the maximum loading steps, production steps, temperature cycles, and adsorption temperature. Additionally, it was indicated that two force fields (COMPASS III and Dreiding) were employed, however, the specific gases utilized and the corresponding force field parameters were not delineated. It is therefore recommended that this information be included in the supporting information.
3. On page 3, line 123, and lines 130-139, the authors state in the text that the GCMC method was used to simulate the adsorption isotherm in a supercell with 10 Å. However, the authors indicate that the cutoff distance set for long-range interactions is 15.5 Å in line 137, and that the cutoff distance should normally be half the length of the cell or even less. This phenomenon should be explained.
4. In Table 4 on page 7 and Fig. 4 on page 11, the DFT simulation results indicated that the C-S framework exhibited the highest adsorption energy for N2O. However, the adsorption amount of N2O was found to be zero in Fig. 4, which contradicted the results in Table 4. The authors must provide a reasonable explanation for this discrepancy.
Author Response
Thank you for your valuable feedback and constructive criticism. We appreciate your time and effort in reviewing our work
Referee 1:
In this manuscript, Vicente Timón and coworkers investigate the copper squarate to absorb a series of small pollutants and some aromatic compounds using DFT calculations in a dry and humid environment. This work demonstrate that water exerts an inhibitory effect on the adsorption of these pollutants, distinguishing copper squarate as a rare MOFs with hydrophilic properties. I suggest this manuscript to be considered as an article in Molecules after taking the following into account:
- On page 3, lines 105-107, the authors mention that a complete geometry optimization was performed for gas molecules, water, and copper squarate. It is recommended that the charge profiles of the optimized gas model and water molecule model, as well as data bond lengths and angles, compared to the reported experimental values. This comparison should be included in the supporting information.
Response: A new table (Table S.1) has been introduced to the Supplementary Information to compare the optimized structures, including the bond lengths and angles, with the corresponding experimental data for the adsorbed gas molecules and water. Furthermore, the following sentence has been included in the first paragraph of Section 3 of the revised manuscript: “The optimized geometries of different species are compared with the corresponding experimental data in Table S1 of supporting information”
- On page 3, lines 109-113, the authors indicate that Monte Carlo simulations were employed to identify the gas adsorption sites and mechanisms within the structure, utilizing the adsorption locator module in Materials Studio. However, they do not provide details regarding the specific settings utilized, including the maximum loading steps, production steps, temperature cycles, and adsorption temperature. Additionally, it was indicated that two force fields (COMPASS III and Dreiding) were employed, however, the specific gases utilized, and the corresponding force field parameters were not delineated. It is therefore recommended that this information be included in the supporting information.
Response:
The parameters used in the Adsorption Locator simulations have been detailed in the text (Methods Section). In addition, at the end, the force field paragraph has also been clarified. The paragraph has been modified as follows (modifications are marked in blue):
After a full geometry optimization of all CH4, CO2, O3, SF6, NF3, CF4, N2O, SO2, 4NP, water, and copper squarate, the most stable adsorption sites of the pollutants inside the solid material were determined in order to minimize the first adsorption energy. The adsorption sites and mechanisms within the copper squarate were determined by Monte Carlo simulations [52], implemented in the Adsorption Locator module [53] using COMPASS III [54] force field in Materials studio 22.0 [55] code. Simulations involves a fraction of 10000 steps following with a number of cycles of 3 with 15000 steps per cycles. Subsequent to the initial results, Density Functional Theory (DFT) calculations were conducted to optimize performance by adjusting the generic force fields within the Adsorption Locator module. More specifically, it was already proved that force fields such as COMPASS III [54] and Dreiding [56] tends to overestimate the number of molecules adsorbed in simulations [57]as was demonstrate in previous studies involving ZIF-4 or ZIF-6 [32,58].
- On page 3, line 123, and lines 130-139, the authors state in the text that the GCMC method was used to simulate the adsorption isotherm in a supercell with 10 Å. However, the authors indicate that the cutoff distance set for long-range interactions is 15.5 Å in line 137, and that the cutoff distance should normally be half the length of the cell or even less. This phenomenon should be explained.
Response: Material studio use characteristic cutoff values, and all the setting parameters are taken generally by defaults. In our case, we note that for the adsorption simulation, we considered the 1×1×2 cell, so the cutoff distance increases automatically.
- In Table 4 on page 7 and Fig. 4 on page 11, the DFT simulation results indicated that the C-S framework exhibited the highest adsorption energy for N2O. However, the adsorption amount of N2O was found to be zero in Fig. 4, which contradicted the results in Table 4. The authors must provide a reasonable explanation for this discrepancy.
Response: The following sentences have been added to the second paragraph of Section 3.3:
As example, N2O shows zero loaded molecules, while the adsorption study shows a high negative adsorption energy of -63.32 kJ/mol, indicating a favorable capture. This aligns also with Connolly studies, which show available space inside copper squarate. These particular results contradict the precise results obtained by the DFT-D3 method. These strange results can be attributed to the limitations of the force field used in this C-S model, which includes pressure and is unable to accurately describe the unusual properties of the material, in particular its negative linear compressibility [37], highlighting the necessity of complementing it with ab-initio calculations. In contrast, water is well described, showing an uptake loading of 15 molecules per cell, which is in good agreement with CASTEP calculations. This agreement may indicate the established hydrophilic nature of the solid. Similarly, ozone molecules show comparable behavior, due to their common nature with water.
Reviewer 2 Report
Comments and Suggestions for Authors
This work explored the adsorption potential of small molecules on a copper squarate MOF by using a theoretically calculated methods, in which the guests’ binding mechanism, binding energy, adsorption isotherms, as well as co-adsorption curves were illustrated. However, I think that it can’t be published before the following concerns are fully considered.
1. The correction method of DFT-D2 is outdated. Why don't you use the DFT-D3? The DFT-D3 has a greater precision than DFT-D2, and does not significantly increase the difficulty of calculations.
2. DFT calculations are not necessarily accurate especial for your sorption isotherms. How do you make sure that the data is trustworthy? Whether there are relevant literatures data or experiments to support results? It is suggested to supplement the individual gas adsorption data for comparison with the calculated adsorption and co-adsorption data.
3. Why did the authors use different probe to estimate the volume and surface area? It is very confusing.
4. Line 226-227, why does the shorter O-H bond in humid conditions indicate the stability of the MOF structure? It is suggested to provide some experimental data, for instance, PXRD patterns after treating the MOF samples in aqueous solution with different pH values for a certain time.
5. In Line 281, Page 8, what is caused by electron density? And there have not any data about electron density in Figure 3 and the full text.
6. How did you get the theoretical density value of copper squarate? Line 167, Page 5.
7. The critical atoms distances between pollutants and MOF should be labeled in Figure 2 and Figure 3.
8. The description of equation is incomplete Line 119-122, Page 3.
9. In equation (3), What does “n-1” stand for?
10. There is a strange symbol in the Line 302, Page 9.
11. There is a writing error, “[2.8473.080] Å”, Line 216, Page 7.
12. Figure 4 should be replaced with a higher resolution picture.
Comments on the Quality of English Language
Minor editing of English language required
Author Response
Thank you for your valuable feedback and constructive criticism. We appreciate your time and effort in reviewing our work
Referee 2:
This work explored the adsorption potential of small molecules on a copper squarate MOF by using a theoretically calculated methods, in which the guests’ binding mechanism, binding energy, adsorption isotherms, as well as co-adsorption curves were illustrated. However, I think that it can’t be published before the following concerns are fully considered.
- The correction method of DFT-D2 is outdated. Why don't you use the DFT-D3? The DFT-D3 has a greater precision than DFT-D2 and does not significantly increase the difficulty of calculations.
Response: Effectively, we made a mistake. During the calculation, we used the visual interface of Material studio, which does not specify the type of Grimme’s dispersion to be employed. Therefore, we assumed it was DFT-D2 based on our previous knowledge. Now, after reading this comment, we research and found that from Material studio 19 and later versions, Grimme’s dispersion correction employed is DFT-D3. We have corrected this in the text of the revised manuscript. One reference concerning DFT-D3 (Ref. [48]) has been included in the revised manuscript: S. Ehrlich, J. Moellmann, W. Reckien, T. Bredow, S. Grimme, System-dependent dispersion coefficients for the DFT-D3 treatment of adsorption processes on ionic surfaces, ChemPhysChem 12 (17) (2011) 3414–3420; https://doi.org/ 10.1002/cphc.201100521.
- DFT calculations are not necessarily accurate especial for your sorption isotherms. How do you make sure that the data is trustworthy? Whether there are relevant literatures data or experiments to support results? It is suggested to supplement the individual gas adsorption data for comparison with the calculated adsorption and co-adsorption data.
Response: Theoretical DFT calculations appear to provide accurate adsorption isotherms. See, for example: C. Moya, D. Hospital-Benito, R. Santiago, J. Lemus, J. Paloma, Prediction of CO2 chemical absorption isotherms for ionic liquid design by DFT/COSMO-RS calculations, Chemical Engineering Journal Advances 4 (2020) 100038; https://doi.org/10.1016/j.ceja.2020.100038. However, force fields are not as accurate as DFT. In the present case, copper squarate has never been subject of any theoretical or experimental adsorption work which can help us to provide some support to the results of the calculations. In addition, sorption isotherm shows some contradictions with ab-initio results and Connolly analysis which are known to be more accurate. This is why it is essential for a trustworthy result to use them after the force-field prediction of sorption module. Particularly since copper squarate is an unusual material with negative linear compressibility.
- Why did the authors use different probe to estimate the volume and surface area? It is very confusing
Response: The probe radius is chosen to estimate the volume and the surface area proper for each pollutants involved. Depending on the surface area, hindrance between functional groups, and the presence of non-bounding electron pairs that create larger electron clouds, different radius values are implied. Therefore, a larger radius is used to account for probable interactions with other molecules and to analyze the volume they occupy, including the entire space surrounding the molecule. Some manuscript enumerates this phenomenon (for example, E. P. F. Nhavene, et al., 10.1016/j.micromeso.2015.06.035). For H2O, SO2, N2O, and CH4, we use a probe radius of 1.37 Å, for 4NP, 1.4 Å, and, finally, for the more electronegative molecules CO2, O3, NF3, SF6, CF4, and 2NP, a larger probe radius of 1.7 Å was chosen.
- Line 226-227, why does the shorter O-H bond in humid conditions indicate the stability of the MOF structure? It is suggested to provide some experimental data, for instance, PXRD patterns after treating the MOF samples in aqueous solution with different pH values for a certain time
Response: It is well known from literature that MOFs are not exceptionally stable at humid conditions. In our case, copper squarate shows negative adsorption energy with OH bond lengths of 0.962 Å and 0.966 Å, which are close to the experimental value of 0.957 Å (http://refhub.elsevier.com/S1387-1811(15)00368-6/sref46) and similar to studies on the adsorption zeolites (0.97 Å for ZIF-6*, 0.94 Å for ZIF-1*, and 0.98 Å for ZIF-4*) concluding a non-distortion of water inside this MOFs.
*El, N.; Timón, V.; Boussessi, R.; Dalbouha, S.; Senent, M.L. DFT Studies of Single and Multiple Molecular Adsorption of CH4, SF6 and H2O in Zeolitic-Imidazolate Framework (ZIF-4 and ZIF-6). Inorganica Chim Acta 2019, 490, 272–281, doi:10.1016/j.ica.2019.03.016
- In Line 281, Page 8, what is caused by electron density? And there have not any data about electron density in Figure 3 and the full text.
Response: The charge density involving the various oxygen atoms of ozone pollutants and copper squarate, along with their non-bonding electron pairs, increases the electronic density creating a big electron cloud, leading to London attractive interaction. For this reason, the adsorption energy is favorable in the case of ozone. Meanwhile, Figure 3 show the co-adsorption of 1 pollutant with 11 water; in this case (line 281), the paragraph is talking about ozone’s co-adsorption.
- How did you get the theoretical density value of copper squarate? Line 167, Page 5.
Response: The theoretical density of copper squarate is obtained from the property results of its fully optimized structure using Material Studio software.
- The critical atoms distances between pollutants and MOF should be labeled in Figure 2 and Figure 3.
Response: These labels have been included in Figures 2 and 3.
- The description of equation is incomplete Line 119-122, Page 3.
Response: Equation (3) and the corresponding descriptive sentence has been changed in the revised manuscript
where represents the total energy of the solid material after adsorption of the pollutants by copper squarate bulk and is the sum of the total energy of the previously adsorbed species calculated separately.
- In equation (3), ¿ What does “” stand for?
Response: Equation (3) has been changed in the revised manuscript as described in the previous comment.
10.There is a strange symbol in the Line 302, Page 9.
Response: The paragraph has been revised.
- There is a writing error, “[2.8473.080] Å”, Line 216, Page 7.
Response: Effectively, yes. This error was corrected to became [2.847-3.080] Å.
- Figure 4 should be replaced with a higher resolution picture.
Response: Figure 4 was replaced by an improved one.
Reviewer 3 Report
Comments and Suggestions for Authors
In this work, the authors demonstrated that water exerts an inhibitory effect on the adsorption of these pollutants, distinguishing copper squarate as a rare MOFs with hydrophilic properties. The Connolly surface was estimated to give a more accurate idea of the volume and surface accessibility of copper squarate, pointing out that copper squarate is an efficient adsorbent for small 26 molecular pollutants and greenhouse gases. Personally, I would like to see this work published on Molecules after some minor revisions, detailed below:
1) In order to make the whole work consistent, it is recommended that the author use the Sorption module to predict the adsorption configuration and adsorption isotherm, since the Adsorption Locate module is more used for surface adsorption rather than pore adsorption.
2)In Figure 4, the adsorption isotherm of CH4 clearly has a singular point, and the authors are advised to rerun the simulations to confirm the reliability of the calculated results.
3)Some relevant literatures, such as Coordination Chemistry Reviews, 2022, 469: 214670; Advanced Science, 2023, 10(21): 2301461; etc. could be used to be borrowed and referenced to improve the simulations.
Comments on the Quality of English Language
Bascially, the language of this paper is accurate and fluent.
Author Response
Thank you for your valuable feedback and constructive criticism. We appreciate your time and effort in reviewing our work
Referee 3:
In this work, the authors demonstrated that water exerts an inhibitory effect on the adsorption of these pollutants, distinguishing copper squarate as a rare MOFs with hydrophilic properties. The Connolly surface was estimated to give a more accurate idea of the volume and surface accessibility of copper squarate, pointing out that copper squarate is an efficient adsorbent for small 26 molecular pollutants and greenhouse gases. Personally, I would like to see this work published on Molecules after some minor revisions, detailed below:
- In order to make the whole work consistent, it is recommended that the author use the Sorption module to predict the adsorption configuration and adsorption isotherm, since the Adsorption Locate module is more used for surface adsorption rather than pore adsorption
Response: The Sorption module is used in our study. In material studio software used in our work, sorption module helps us to predict adsorption isotherms to learn more about the number of molecules insert inside the material. For the configuration of the most stable adsorption sites, we used the Adsorption locator module following CASTEP calculations. This methodology has already been used and validated in the adsorption studies of porous materials such as ZIFs. Here are the links:
- El, N.; Timón, V.; Boussessi, R.; Dalbouha, S.; Senent, M.L. DFT Studies of Single and Multiple Molecular Adsorption of CH4, SF6 and H2O in Zeolitic-Imidazolate Framework (ZIF-4 and ZIF-6). Inorganica Chim Acta 2019, 490, 272–281, doi:10.1016/j.ica.2019.03.016.
- Timón, V.; Senent, M.L.; Hochlaf, M. Structural Single and Multiple Molecular Adsorption of CO2 and H2O in Zeolitic Imidazolate Framework (ZIF) Crystals. Microporous and Mesoporous Materials 2015, 218, 33–41, doi:10.1016/j.micromeso.2015.06.035.
- In Figure 4, the adsorption isotherm of CH4 clearly has a singular point, and the authors are advised to rerun the simulations to confirm the reliability of the calculated results
Response: Effectively, Figure 4 provides a prediction about the amount of CH4 adsorbed. Meanwhile, the two previous sections prove the ability of copper-squarate to capture CH4. First, the Connolly study confirms that there is enough space inside the pore, and the adsorption section confirms this with a negative adsorption energy using the accurate DFT method.
- Some relevant literatures, such as Coordination Chemistry Reviews, 2022, 469: 214670; Advanced Science, 2023, 10(21): 2301461; etc. could be used to be borrowed and referenced to improve the simulations.
Response: These references have been included (Refs. [34] and [43]).
Round 2
Reviewer 1 Report
Comments and Suggestions for Authors
The authors have revised the paper so that it can now be recommended for publication.
Author Response
Thank you for your valuable feedback and constructive criticism
Referee 1:
We would like to inform you that, following the indications of one of the referees, we have improved the Powder X-ray Diffraction (section 3) of the manuscript. We have included a Powder X-ray Diffraction (PXRD) pattern of Copper Squarate MOF derived from the computed structure, compared with the pattern derived from the experimental structure in Figure S.1, which we have added to the Supplementary Information. Accordingly, a sentence has been included in the first paragraph of Section 3.
Reviewer 2 Report
Comments and Suggestions for Authors
For Comments 2 and 3, the experimental gas adsorption and PXRD data were still not provided, so i am not sure that the theoretical results are reliable.
Author Response
Thank you for your valuable feedback and constructive criticism. We appreciate your time and effort in reviewing our work again.
Referee 2:
For Comments 2 and 3, the experimental gas adsorption and PXRD data were still not provided, so I am not sure that the theoretical results are reliable.
Response: The Powder X-ray Diffraction (PXRD) pattern of Copper Squarate MOF derived from the computed structure is compared with the pattern derived from the experimental structure in Figure S.1, which we have added to the Supplementary Information. As can be seen in this figure, the patterns are consistent thus confirming the accuracy of the computed structure. The following sentence has been included in the first paragraph of Section 3.
“The Powder X-ray diffraction (PXRD) patterns of copper squarate was derived from the computed and experimental structures using program REFLEX included in Materials Studio program suite [55]. The results are displayed Figure 1 of the Supplementary Material and as can be seen they are consistent.”
Furthermore, Copper Squarate has recently been synthesized and analyzed using PXRD by us and our collaborators. The results will be published in a forthcoming paper. Again, the experimental and theoretical patterns are in excellent agreement.
Copper squarate has never been subject of any theoretical or experimental adsorption work and, therefore, we cannot currently provide some experimental data to support to the results of the isothermal calculations. As mentioned in the previous paragraph, Copper Squarate has recently been synthesized and experimental adsorption isotherms will be measured, so that the results may be compared with the theoretical ones and its accuracy evaluated.
